# RE-CALIBRATING PROGRESS: A PHYSICS-AWARE BENCHMARK TO EXPOSE THE EVALUATION GAP IN SCIENTIFIC MACHINE LEARNING

## ABSTRACT

Progress in scientific machine learning is critically hindered by a pervasive "evaluation gap", where models that excel on legacy benchmarks fail in real-world deployment due to a reliance on idealized synthetic data and fragile proxy metrics. We argue that the path forward requires a new paradigm of physics-aware benchmarking, which we instantiate with PRISMABENCH for the challenging inverse problem of hyperspectral pansharpening. Our ecosystem introduces three core contributions: a **physics-enriched dataset** that packages real satellite PRISMA hyperspectral (HS) and panchromatic (PAN) pairs by their real physical sensors with 10 challenge scenes; an extended **PAN-centric evaluation metric**, including a novel physics-consistency score for robust, no-reference assessment; and **insightful visualization tools**, such as multi-metric radar charts, to move beyond single-score leaderboards and expose performance trade-offs. Using this framework, we reveal a critical disconnect: a model's rank on traditional reduced resolution benchmarks is a limited predictor of its real-world performance. By open-sourcing our ecosystem, we provide a blueprint for creating benchmarks that challenge the community to move beyond optimizing flawed proxies and towards developing models that are demonstrably robust and physically plausible.

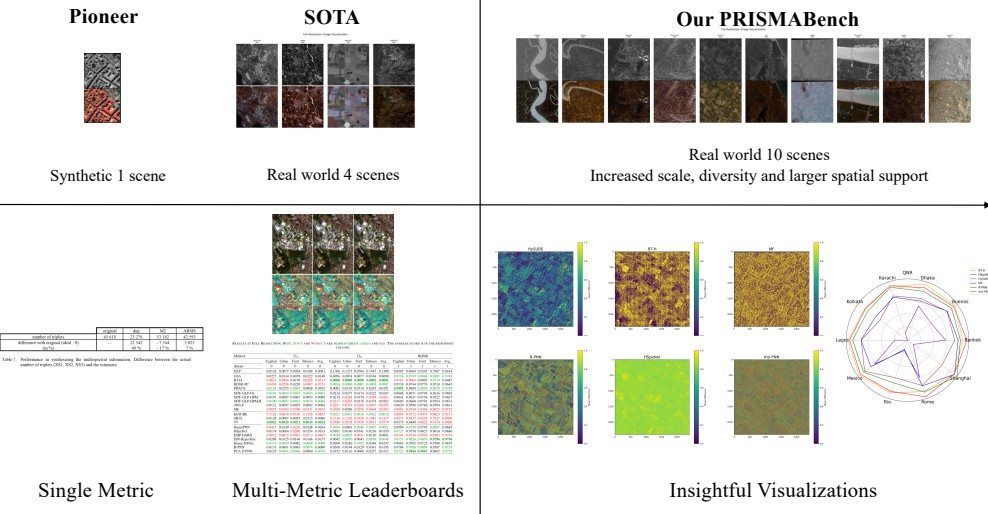

Figure 1: **From Misleading Leaderboards to Actionable Insight**. (**Left**) A traditional single-score / multi-metric leaderboards mask performance trade-offs, sometimes suggesting one model is uniformly superior. (**Right**) Our benchmark advocates for a shift to insightful visualizations, such as this multi-metric radar chart. This profile immediately reveals that the top-ranked model is actually a "specialist" that fails on certain criteria, while another model offers more balanced, "generalist" performance. This holistic view provides a more reliable and actionable basis for model selection.

# 1 INTRODUCTION

High-resolution hyperspectral imagery (HSI) underpins applications from precision agriculture to climate monitoring (Pooja Vinod Janse, 2017; Van der Meer et al., 2012). Yet a fundamental photon–resolution trade-off in orbital sensing prevents any single platform from jointly delivering meter-scale detail and dense spectra (Schowengerdt, 2006). *Hyperspectral pansharpening* (HS-PAN) addresses this limitation by fusing a low-resolution hyperspectral cube (LR-HSI) with a high-resolution panchromatic image (HR-PAN) to synthesize a high-resolution hyperspectral product (HR-HSI).

**Why evaluation is hard.** Hyperspectral pansharpening (Bertero & Boccacci, 1998) is a classic *ill-posed inverse problem*: To understand the key challenge in pansharpening, it is important to first consider the short-wave infrared (SWIR) spectrum. SWIR refers to a range of light invisible to the human eye, which is useful for identifying materials like minerals, soil moisture, and vegetation types. However, most standard high-resolution panchromatic (PAN) sensors have no sensitivity in this range. Therefore, many HR-HSI solutions are consistent with the same LR-HSI/HR-PAN pair, and the inverse is unstable to noise, atmospheric effects, and sub-pixel misregistration. Two factors exacerbate this: (i) **non-uniqueness**, because PAN conveys edge structure but little spectral identity, especially in SWIR where PAN overlap is weak; and (ii) **instability**, where small perturbations can amplify into spectral distortion or spatial artifacts. Consequently, what constitutes a "good" reconstruction depends as much on the *evaluation protocol* as on the model itself.

**Legacy protocols create an evaluation gap.** The community predominantly measures progress using (a) Wald-style reduced-resolution (RR) tests that synthetically degrade a reference image (Wald, 1999), and (b) full-resolution (FR) no-reference indices (e.g., QNR and components) when no ground truth exists. Both have well-documented limits. First, **spectral mismatch**: PAN typically spans VIS–NIR while HSI extends to SWIR; naive consistency checks penalize (or ignore) bands for which PAN conveys no reliable guidance (Vivone et al., 2014; Ciotola et al., 2024). Second, a **synthetic→real gap**: RR degradations omit real sensor noise, PSF/SRF mismatch, and misregistration, so RR gains can fail to translate to FR performance (Ciotola et al., 2024). Third, **brittle FR metrics**: no-reference scores can be "gamed" (e.g., over-sharpening vs. spectral fidelity) and correlate weakly with perceived and physically plausible quality (Arienzo et al., 2022). Together, these issues leave an *evaluation gap*: leaderboard wins under idealized setups do not guarantee robustness under real acquisition physics (Fig. 2).Traditional evaluation methods fail due to fundamental physical and procedural flaws.

**Our Contributions: A Framework for Realistic Evaluation.** To re-calibrate progress towards real-world utility, we introduce PRISMABENCH, a comprehensive ecosystem designed to address the core evaluation gaps. Our framework is built on three synergistic contributions:

1. **A Large, Diverse, and Challenging Dataset:** We introduce the PRISMABENCH dataset, a new, large-scale collection of over 10 globally distributed PRISMA scenes. It addresses the limitations of prior benchmarks by providing greater geographical diversity for robust generalization testing and by utilizing larger image tiles to preserve the fine-grained spatial structures essential for realistic evaluation. Furthermore, we introduce a novel data richness score to ensure our test set is demonstrably complex and challenging.

2. **Principled Metrics for Reliable Evaluation:** To overcome the well-documented fragility of legacy no-reference scores, we propose the **PAN-Conditioned Spatial Score** ($D_\rho^{\mathbf{PAN}}$). This physics-aware metric measures spatial consistency *only* on the subset of hyperspectral bands that physically overlap with the panchromatic sensor's response. This targeted approach provides a more robust and less gameable measure of true spatial detail transfer than traditional, full-spectrum metrics.

3. **A Modernized Toolbox with Insightful Visualizations:** We provide a standardized, open-source toolbox that, for the first time, includes strong baselines from self-supervised and generative paradigms. Crucially, our framework moves beyond single-score leaderboards by standardizing insightful visualization tools, such as **multi-metric radar charts**, which provide a holistic profile of a model's behavior and expose the critical performance trade-offs that aggregate scores conceal.

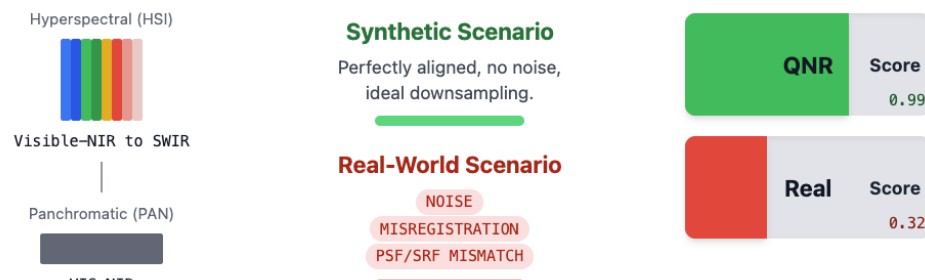

Figure 2: **Why Pansharpening Benchmarking Fails under Real-World Conditions.** The three core challenges of the pansharpening task systematically undermine legacy evaluation protocols. (**a**) *Spectral Mismatch* between the PAN and HSI sensors makes evaluating spectral fidelity in the SWIR bands unreliable. (**b**) The *Synthetic-to-Real Gap*, caused by reliance on idealized Wald protocol operators, means performance on benchmarks often fails to predict real-world utility. (**c**) *Fragile No-Reference Metrics* can be "gamed" by algorithms, creating a disconnect between high scores and true visual quality.

## 2 PRELIMINARIES: THE PANSHARPENING INVERSE PROBLEM

Hyperspectral pansharpening is a classic ill-posed inverse problem central to remote sensing. It aims to reconstruct a high-resolution hyperspectral image ($\widehat{X}_{\mathrm{HR}} \in \mathbb{R}^{H \times W \times L}$) by fusing a low-resolution hyperspectral cube ($H_{\mathrm{LR}} \in \mathbb{R}^{h \times w \times L}$) with a high-resolution panchromatic image ($P_{\mathrm{HR}} \in \mathbb{R}^{H \times W \times 1}$).

The task is governed by the sensor's physical forward model. An ideal reconstruction $\widehat{X}_{\mathrm{HR}}$ must simultaneously satisfy two physical consistency constraints: (1) when spatially degraded by the sensor's point-spread function ($\mathcal{B}$) and downsampler ($\mathcal{D}$), it must match the observed LR-HSI; and (2) when spectrally integrated by the sensor's spectral response function ($\mathcal{R}$), it must match the observed HR-PAN. This can be expressed as:

$$\mathcal{D}\big(\mathcal{B}(\widehat{X}_{\mathrm{HR}})\big) \approx H_{\mathrm{LR}} \quad \text{and} \quad \mathcal{R}(\widehat{X}_{\mathrm{HR}}) \approx P_{\mathrm{HR}}. \tag{1}$$

The ill-posed nature of this problem, coupled with the absence of a real-world ground truth, dictates that evaluation must fundamentally differ from standard supervised tasks. To crystallize these differences, Table 1 contrasts the core objectives and methodologies of pansharpening against common downstream analysis tasks like classification and unmixing.

Table 1: Task positioning. Pansharpening is a physics-constrained inverse problem focused on image reconstruction, which is fundamentally distinct from downstream analysis tasks that optimize for different scientific objectives and employ different evaluation metrics.

| Dimension | Pansharpening (Fusion) | Classification (Analysis) | Unmixing (Analysis) |
|---|---|---|---|
| **Primary Goal** | Reconstruct a high-resolution image | Assign a semantic label | Estimate sub-pixel material abundances |
| **Supervision** | No ground truth on real data; self-consistency | Per-pixel categorical labels | Often unsupervised or semi-supervised |
| **Core Metric** | Physics consistency (e.g., Q2n, QNR) | Classification accuracy (e.g., OA, $\kappa$) | Error (e.g., RMSE, MAE) |

## 3 RELATED WORK

The evaluation of hyperspectral pansharpening is built upon two legacy protocols in multispectral pansharpening, each with well-documented limitations. The seminal (Wald, 1999)'s protocol enables objective, reference based reduced-resolution (RR) evaluation to generate synthetic data from high-resolution hyperspectral images like Pavia (Gamba & Dalponte, 1999), Botswana (NASA, 2001), but creates a persistent synthetic-to-real domain gap. For real, full-resolution (FR) data, the field relies on **no-reference metrics** like QNR (Alparone et al., 2008), which have been shown to be fragile and poorly correlated with physical plausibility, especially in the challenging SWIR bands (Arienzo et al., 2022). As shown in Table 2, we observed a disconnect between the two metrics.

Table 2: PRISMA toolbox (Ciotola et al., 2024) benchmark performance in reduced-resolution (RR) and full-resolution (FR). **Bold** indicate the best.

| Method | ERGAS | SAM | $Q2^n$ (**RR**) | $D_\lambda$ (**FR**) | $D_s$ (**FR**) | $QNR$ (**FR**) |
|---|---|---|---|---|---|---|
| Ideal | 0 | 0 | 1 | 0 | 0 | 1 |
| **PRACS** | 2.4461 | 3.7174 | 0.7723 | 0.0102 | **0.0151** | **0.9749** |
| **AWLP** | 2.7697 | 5.2381 | 0.7783 | 0.0094 | 0.0295 | 0.9613 |
| **HSpeNet** | 1.7379 | **3.3525** | **0.8660** | 0.0163 | 0.0159 | 0.9680 |
| **R-PNN** | **1.7356** | 3.4026 | 0.8555 | **0.0090** | 0.0195 | 0.9716 |

The top-performing model HSpeNet under the RR metric failed to be the best in the FR evaluation, whereas a different model PRACS with a modest RR score achieved one of the top results on the FR metric. While recent efforts have standardized the application of these protocols (Vivone et al., 2021), the underlying methodological flaws remain, creating a critical need for a more realistic evaluation framework.

Our work is motivated by the need to bridge this evaluation gap by rigorously testing modern machine learning paradigms that are well-suited to this task. Hyperspectral pansharpening is a classic **ill-posed inverse problem**, a domain where generative models like Denoising Diffusion Probabilistic Models (DDPMs) have shown immense promise (Ho et al., 2020; Song et al., 2021). The lack of labeled data also makes **self-supervised learning (SSL)**, particularly with the advent of Earth observation foundation models like SatMAE (Cong et al., 2022), a critical enabler. Finally, the explicit physical nature of the problem calls for **physics-informed** approaches that embed sensor constraints directly into the learning process (Karniadakis et al., 2021).

However, these three powerful ML paradigms—generative priors, SSL features, and physics-informed learning—have been largely developed in isolation from the specific realities of pansharpening. It remains an open question whether these methods are robust to real-world conditions like sensor mismatch and misregistration. With PRISMABENCH, we aim to provide a unified platform to begin addressing this question, offering a physics-enriched dataset, robust metrics, and modern baselines for a fair and rigorous comparison.

## 4 PRISMABENCH: A PHYSICS-AWARE EVALUATION ECOSYSTEM

To address the critical evaluation gap in hyperspectral pansharpening, we introduce PRISMABENCH, a comprehensive ecosystem designed to re-calibrate research towards real-world utility and physical plausibility. Our framework is built on three synergistic pillars: (1) a large-scale, **physics-enriched dataset** that provides the ground-truth sensor operators; (2) a suite of **principled evaluation metrics**, including novel physics-consistency and robustness scores; and (3) a set of **insightful visualization tools** designed to move beyond single-score leaderboards and reveal deeper performance trade-offs.

### 4.1 PILLAR 1: A PHYSICS-ENRICHED DATASET

Our benchmark, PRISMABENCH, is built upon a new, large-scale PRISMA dataset that significantly enhances the scale and rigor of hyperspectral pansharpening evaluation (Table 3). Its advantages are threefold:

- **Increased Scale and Diversity:** With 10 globally distributed test scenes, our benchmark provides greater statistical power and a more robust test of model generalization compared to previous datasets, which were often limited to a few specific geographies. The selection of the scenes from 1500+ downloaded PRISMA imagery [1] are based on the richness score (see Appendix A.2).

- **Larger Spatial Support:** State-of-the-art (SOTA) benchmarks typically evaluate performance on small patches, ranging from $120\times120$ to $2400\times2400$ pixels, which are cropped from large PRISMA PAN images ($6000\times6000$). In contrast, we use large image tiles ($2400\times2400$) to better preserve the fine-grained structures that are critical for realistic, full-resolution stress testing.

---

[1]https://prisma.asi.it/js-cat-client-prisma-src/

This physics-enriched data underpins our novel evaluation metrics, including a PAN-conditioned spatial score ($D_\rho^{\text{PAN}}$), making our benchmark more discriminative, reproducible, and aligned with real-world performance.

Table 3: Comparison of our PRISMABENCH test set with prior corpora used in hyperspectral pan-sharpening research at Full Resolution using PRISMA satellite.

| Benchmark | # Test Scene(s) | Coverage | # Bands (used / full) | Dimension |
|---|---|---|---|---|
| Pavia Center (Gamba & Dalponte, 1999) | 1 | Italy | 102 / 115 | P: 160x160, HS: 40×40 |
| Botswana (NASA, 2001) | 1 | Botswana | 145 / 242 | P: 120x120, HS: 40x40 |
| PRISMA FR (Vivone et al., 2022) | 2 | Italy | 69 / 239 | P: 2400x2400, HS: 400x400 |
| PRISMA toolbox (Ciotola et al., 2024) | 4 | Italy, US, Mexico | 159 / 239 | P: 1200x1200, HS: 200x200 |
| **PRISMABENCH (Ours)** | 10 | Global | 159 / 239 | P: 2400x2400, HS: 400x400 |

## 4.2 PILLAR 2: MOVING BEYOND PROXIES WITH PRINCIPLED METRICS

We move beyond legacy metrics by introducing a protocol that measures what truly matters: physical consistency and robustness to real-world perturbations.

**PAN-Conditioned Spatial Score ($D_\rho^{\text{PAN}}$) for Robust Spatial Evaluation:** A primary failure mode of legacy no-reference evaluation is the unreliability of spatial quality scores. To address this, we propose $D_\rho^{\text{PAN}}$, a new metric grounded in the principle that the high-resolution panchromatic (HR-PAN) image is the most reliable source of ground-truth spatial structure.

**Mechanism:** Unlike traditional $D_\rho$ (Guarino et al., 2025) that perform a naive global consistent comparison, our score is PAN-centric. We define $D_\rho^{\text{PAN}}$ as the spatial consistency between the high-frequency components of the generated HSI within the panchromatic (PAN) range and the corresponding components of the high-resolution PAN (HR-PAN) image, as formulated below:

$$D_\rho^{\text{PAN}} = D_\rho(Y_{:,i:j,:,:}, I_{PAN}), \tag{2}$$

where i and j are the indices representing the spectral range of the PAN sensor (e.g., 400–700 nm for the PRISMA sensor) (Cogliati et al., 2021), Y is the model fused image and $I_{P}AN$ is the original input PAN image as reference. The effectiveness of a model's PAN injection can be measured by the spatial consistency within the PAN range.

**Benefit:** Bands with high spectral overlap (e.g., in the visible range) are strongly expected to match the PAN's structure, and deviations are penalized accordingly. This makes $D_\rho^{\text{PAN}}$ a more physically plausible measure of spatial fidelity.

## 4.3 PILLAR 3: INSIGHTFUL VISUALIZATIONS BEYOND LEADERBOARDS

Numerical scores often fail to capture the nuanced performance trade-offs and failure modes of complex models. A core contribution of PRISMABENCH is therefore a suite of standardized, insightful visualization tools designed to move evaluation beyond simple leaderboards towards a deeper, more holistic understanding of model behavior.

We introduce two key visualization methodologies. First, to diagnose a model's fusion strategy, we propose the **Mean Difference Heatmap** (Fig. 4). This tool visualizes the per-pixel radiometric change a model imparts on the input, providing an immediate "fingerprint" of whether its approach is spectrally conservative or spatially aggressive. Second, to overcome the limitations of single-score rankings, we standardize the use of **Multi-Metric Radar Charts** (Fig. 5). This visualization plots a model's performance across multiple competing criteria (e.g., performance on different scenes, or metrics for quality vs. robustness). It instantly reveals a model's performance profile, clearly distinguishing well-balanced "generalist" models from "specialist" models that excel on one axis at the expense of others.

These tools provide a more complete and actionable picture of a model's true capabilities. A detailed summary of all our proposed visualization methods and their underlying mechanisms, including VIS vs. Invisible band analysis and false-color mapping, is provided in Appendix A.3.

## 5 EXPERIMENTS: EXPOSING THE EVALUATION GAP

Our experiments are designed not merely to rank models, but to rigorously validate our central thesis: that legacy evaluation protocols are fundamentally flawed, and our physics-aware benchmark provides a more reliable measure of real-world performance. We structure our validation around three key empirical findings.

### 5.1 EXPERIMENTAL SETUP

**Datasets and Baselines.** All experiments are conducted on our PRISMABENCH dataset. We compare a wide range of models, including classical (HySURE, MF, BT-H), supervised SoTA (HSpeNet), and unsupervised PNN-based SoTA (R-PNN and $\rho$-PNN).

Table 4: Representative pansharpening methods included in our benchmark. We select strong baselines spanning classical model-based paradigms and both supervised and unsupervised deep learning approaches to ensure a comprehensive and fair comparison.

| Name | Reference | Summary |
|---|---|---|
| HySURE | (Simões et al., 2015) | Bayesian estimation with vector total variation prior. |
| MF | (Restaino et al., 2016) | Nonlinear decomposition with morphological filters. |
| BT-H | (Lolli et al., 2017) | Brovey transform with haze correction. |
| HSpeNet | (He et al., 2020) | Advanced version of HyperPNN (He et al., 2019). |
| R-PNN | (Guarino et al., 2024) | Bandwise pansharpening using modified. Z-PNN with tuning propagation |
| $\rho$-PNN | (Guarino et al., 2025) | Bandwise pansharpening using modified Z-PNN with hysteresis-inspired strategy. |

**Evaluation in Full-Resolution (FR).** We evaluate all models on real, full-resolution satellite imagery to probe the synthetic-to-real gap. Performance is measured with our proposed no-reference metrics: the ($D_\rho^{\text{PAN}}$), alongside the legacy QNR for comparison. As shown in Table 5, the performance of all models dropped compared to SOTA benchmark (Guarino et al., 2025). Although HSpeNet get the best $D_s$, it performed badly on other metrics as well as the visualization in Fig. 4.

Table 5: Comparison of different methods on five metrics. **Bold** indicate the best.

| Method | $D_\lambda$ | $D_s$ | $QNR$ | $D_\rho$ | $D_\rho^{\text{PAN}}$ |
|---|---|---|---|---|---|
| Ideal | 0 | 0 | 1 | 0 | 0 |
| *Traditional methods* | | | | | |
| HySURE | 0.1557 | 0.0046 | 0.8404 | 0.4270 | 0.2534 |
| MF | 0.0776 | 0.1082 | 0.8238 | 0.1271 | **0.1111** |
| BT-H | 0.0652 | 0.0043 | 0.9307 | 0.1665 | 0.1706 |
| *Deep Learning methods* | | | | | |
| HSpeNet | 0.3320 | **0.0001** | 0.6679 | 0.9223 | 0.8687 |
| R-PNN | 0.0337 | 0.0349 | **0.9326** | 0.1403 | 0.2483 |
| $\rho$-PNN | **0.0238** | 0.0737 | 0.9040 | **0.1171** | 0.1201 |

### 5.2 FINDING 1: LEGACY BENCHMARKS ARE LIMITED PREDICTORS OF REAL-WORLD PERFORMANCE

A critical finding from our analysis suggests that a model's success on the traditional, synthetic Reduced-Resolution (RR) benchmark may be a poor predictor of its performance on real Full-Resolution (FR) data. This apparent disconnect, often termed the "synthetic-to-real gap," could represent a significant obstacle to progress, as it raises the possibility that the community may be optimizing for a flawed objective.

**Evidence from Rank Correlation Collapse.** To quantify this gap, we conducted a rigorous meta-analysis using the comprehensive experimental data from the recent benchmark study by (Guarino et al., 2025), specifically the average scores reported in their Tables VII and VIII. We compared the performance rankings of 21 state-of-the-art methods on the RR benchmark (using the standard ER-GAS metric) against their rankings on the FR benchmark (using the spectral fidelity metric $D_\lambda$). As shown in Figure 3, the result is a near-complete collapse of rank correlation. We found a Spearman's rank correlation coefficient of only $\rho = 0.22$, which is not statistically significant ($p > 0.05$).

This profound disagreement reveals critical insights. For instance, the classic **TV** method, which ranks poorly on the synthetic RR test (13th), is the top performer on the real FR data (1st) due to its strong spectral preservation. Conversely, the powerful deep learning model **Hyper-DSNet**, which is

the top-ranked model on the RR benchmark, falls to 8th place on the FR benchmark. This provides direct, quantitative evidence that optimizing for the idealized conditions of the Wald protocol does not guarantee, and may even hinder, the development of models that are robust to the complexities of real-world physics.

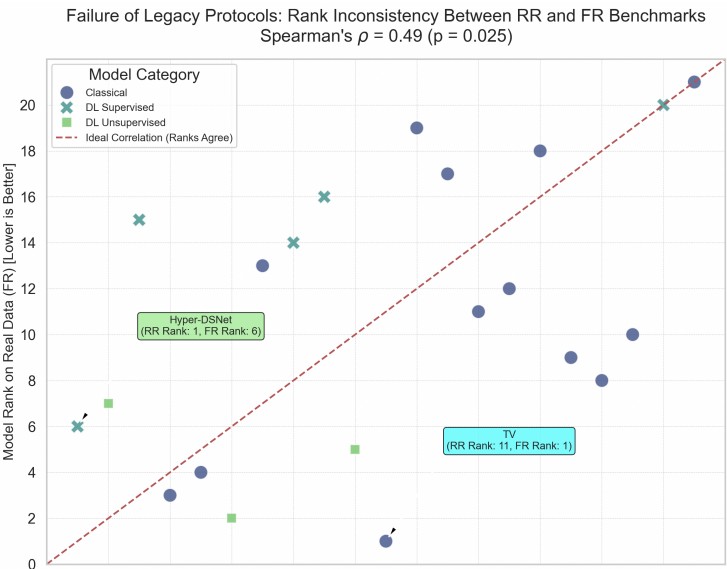

Figure 3: **Failure of Legacy Protocols: A Collapse in Rank Correlation.** Scatter plot of model ranks on the synthetic Reduced-Resolution (RR) benchmark ERGAS versus the real Full-Resolution (FR) benchmark $D_\lambda$. The near-zero rank correlation ($\rho = 0.22$) provides direct quantitative evidence that performance on synthetic data is a poor predictor of real-world performance. This disconnect is starkly illustrated by models like **TV**, which is the top performer on real FR data but ranks poorly on the synthetic benchmark. Data for this analysis is sourced from the comprehensive study by (Guarino et al., 2025).

### 5.3 FINDING 2: PHYSICS-AWARE METRICS REVEAL DEEPER INSIGHTS

Our new visualization tools, grounded in physics and data, expose performance characteristics that are invisible to traditional metrics. The **Mean Difference Heatmap** (Fig. 4) provides a clear "finger-print" of each model's fusion strategy. It visually confirms that aggressive, injection-based methods like BT-H and MF pervasively alter the image's original radiometry (widespread yellow areas), risking spectral distortion. In contrast, more conservative methods like HySURE make targeted, localized adjustments, better preserving spectral integrity.

Crucially, our **VIS vs. Invisible Band Analysis** (see Appendix Fig. 6) allows us to diagnose performance in the challenging SWIR bands where PAN guidance is absent. We found that while most models can reconstruct VIS bands reasonably well, many fail to preserve the unique spectral signatures of features like waterways in the SWIR range. This analysis provides a critical diagnostic tool for assessing the scientific reliability of a model, a dimension completely missed by aggregate metrics.

### 5.4 FINDING 3: MANY MODELS FAIL TO PRESERVE CRITICAL NON-VISIBLE SPECTRAL INFORMATION.

Aggregate spectral metrics can be dangerously misleading, as they often average out poor performance in the scientifically critical short-wave infrared (SWIR) range with good performance in the PAN-overlapping visible (VIS) range. Our VIS vs. Invisible Band Analysis provides a powerful diagnostic tool to expose this failure mode.

As shown in Figure 6, we deconstruct a model's output by separately averaging its VIS and SWIR bands and computing their difference. The difference map highlights features, such as waterways,

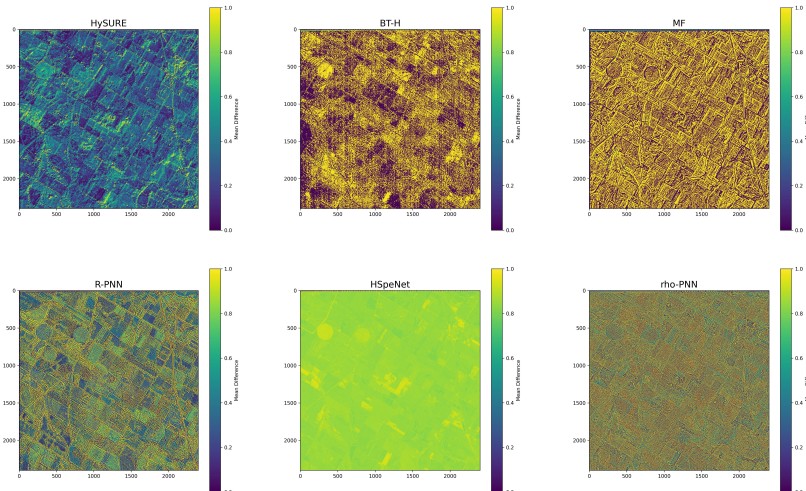

Figure 4: **Visualizing the Fusion Strategy: A Diagnostic Heatmap for Model Behavior.** This figure introduces the Mean Difference Heatmap as a tool to visualize the fundamental trade-off between spatial detail injection and spectral preservation. Each panel plots the per-pixel mean difference between the input and output spectra; hotter colors (yellow) indicate aggressive radiometric alteration, while cooler colors (blue/green) signify a conservative approach that prioritizes spectral fidelity. The analysis clearly distinguishes aggressive methods like **MF** and **BT-H**, which globally alter the input, from conservative methods like **HySURE** and $\rho$**-PNN**, which better preserve the original information. This qualitative "fingerprint" of a model's strategy provides a crucial insight that aggregate numerical scores fail to capture.

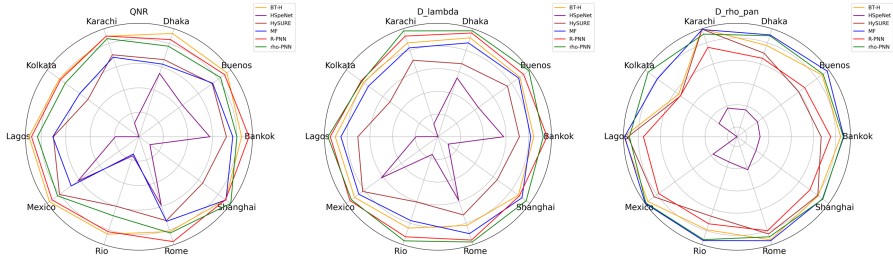

Figure 5: **Beyond Leaderboards: Holistic Performance Profiling with Radar Charts on 10 new challenging scenes.** This radar chart exposes the limitations of single-score rankings by visualizing model performance across multiple evaluation criteria (four test scenes and their average). Each colored line represents a different method's performance profile. The visualization immediately reveals critical performance trade-offs: methods like **MF** (blue line) emerge as "specialists" that excel on specific scenes, but are suboptimal elsewhere. In contrast, methods like **R-PNN** and $\rho$**-PNN** are "generalists" with more balanced, though not always peak, performance. This holistic view provides a much richer and more actionable understanding of a model's true capabilities than a single, aggregated score.

that have a unique and strong signature in the SWIR spectrum. Our analysis reveals that while most models can plausibly reconstruct the VIS bands, many fail to preserve the distinct structure of the SWIR bands, resulting in a low-contrast, washed-out difference map. In contrast, top-performing, physics-aware models successfully maintain a strong contrast, providing direct visual evidence of their superior spectral fidelity in bands where the panchromatic image offers no guidance. This demonstrates that our visualization is a crucial tool for assessing the scientific reliability of a pan-sharpening method.

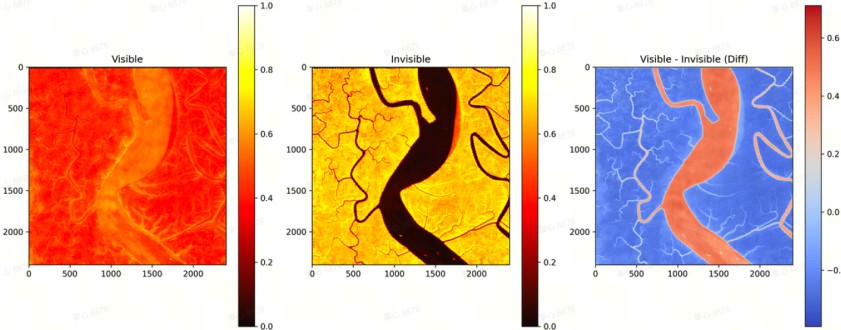

Figure 6: **Diagnosing Spectral Fidelity in Non-Visible Bands.** This visualization deconstructs a model's performance. (**Left**) The model output R-PNN at scene Kolkata averaged over the VIS bands. (**Middle**) The output averaged over the SWIR bands. (**Right**) The difference map, which highlights features with unique SWIR signatures. A strong contrast in this map, as seen here, indicates the successful preservation of scientifically critical, out-of-band information.

## 6 DISCUSSION

Our work is motivated by the observation that in scientific machine learning, the evaluation protocol often shapes the direction of research. We found that the long-standing "evaluation gap" in hyperspectral pansharpening may represent a significant barrier to progress. In response, we developed PRISMABENCH to explore a new approach to evaluation grounded in physical reality, and our findings may have implications for both the remote sensing and broader ML communities.

**Implications for the Remote Sensing Community.** Our results suggest a potential path forward from the difficult trade-off between synthetic reference metrics and fragile no-reference scores. We propose that the community consider adopting our **PAN-Conditioned Spatial Score** ($D_\rho^{\text{PAN}}$) as tools for no-reference evaluation. Because these scores are tied to the sensor's known physics, they appear to be more robust and less susceptible to gaming, potentially helping to re-align the field's objective towards physical plausibility and verifiable spatial fidelity.

**A Potential Blueprint for Scientific Benchmarking.** While instantiated in remote sensing, the principles of PRISMABENCH may offer a portable blueprint for building more reliable benchmarks in other scientific domains. We suggest two key ideas for consideration: (1) enriching datasets by packaging them with their known physical forward operators, and (2) quantifying model reliability directly via stress tests that measure performance under non-ideal conditions. We hope this approach can help foster the development of models that are not just accurate, but also trustworthy.

**Limitations.** We readily acknowledge that our work is a first step. Challenges like severe atmospheric effects and non-static misregistration remain open problems. While our benchmark provides the infrastructure to begin studying these effects, developing models that are fully robust to them is a significant challenge for future research.

## 7 CONCLUSION

In this work, we explored a critical "evaluation gap" where a disconnect between legacy benchmarks and real-world physics may be hindering progress in a key scientific domain. We introduced PRISMABENCH, a physics-aware benchmark ecosystem designed to help re-calibrate research towards verifiable physical consistency and operational robustness. By providing a large-scale dataset, a suite of principled evaluation metrics, and a modernized toolbox, we hope to have created a useful foundation for the next generation of pansharpening research. We release all artifacts to the community to foster a more transparent, reproducible, and impactful era of scientific machine learning.

## REPRODUCIBILITY STATEMENT

All experiments were conducted on a server equipped with 1 NVIDIA 3090 GPU using the provided Docker container, which is based on PyTorch 2.1, CUDA 11.8, and Python 3.10. All random seeds were fixed across runs to ensure deterministic behavior. The complete code, data, and artifacts required to reproduce all findings will be made publicly available.

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

## A  APPENDIX

### A.1  SCENE LIST

The following table presents 10 challenging scenes from various locations worldwide of PRISMA imagery. You can search at their platform to download the original picture.[2]

Table A.6: Summary of 10 PRISMA Satellite Scenes

| Scene Location | Acquisition Date | Start Time (UTC) | End Time (UTC) | Predominant Land Features |
|---|---|---|---|---|
| Kolkata | 2020-12-15 | 04:46:55 | 04:46:59 | Urban, Riverine |
| Dhaka | 2021-10-30 | 04:45:26 | 04:45:30 | Urban, Riverine |
| Bangkok | 2021-11-19 | 03:54:14 | 03:54:19 | Urban, Coastal, Riverine |
| Mexico City | 2022-01-06 | 17:16:58 | 17:17:02 | Urban, Valley |
| Buenos Aires | 2022-02-15 | 14:10:02 | 14:10:07 | Urban, Coastal |
| Rio de Janeiro | 2022-02-23 | 13:06:47 | 13:06:51 | Urban, Coastal, Mountainous |
| Lagos | 2023-12-24 | 10:18:06 | 10:18:10 | Urban, Coastal, Lagoon |
| Shanghai | 2024-08-03 | 02:49:00 | 02:49:04 | Urban, Coastal, River Delta |
| Rome | 2024-08-12 | 10:08:07 | 10:08:11 | Urban, Riverine |
| Karachi | 2024-10-01 | 06:21:54 | 06:21:59 | Urban, Coastal, Arid |

### A.2  SCENE SELECTION BY RICHNESS SCORE

The Richness Score ($S_R$) for a given satellite scene is calculated as the mean of four normalized metrics that quantify its spectral and spatial information content. The formula is as follows:

$$S_R = \frac{1}{4}\left(\frac{V_\lambda}{\max(V_\lambda)} + \frac{H_\lambda}{\max(H_\lambda)} + \frac{H_P}{\max(H_P)} + \frac{\sigma_P}{\max(\sigma_P)}\right) \tag{3}$$

If a scene is identified as urban and the `urban_focus` option is enabled, the score is multiplied by a boost factor ($B_U$):

$$S_{R,\text{urban}} = B_U \times S_R \tag{4}$$

- $S_R$: The final Richness Score for a scene. This is referred to as `composite_score` in the Python script.

- $V_\lambda$: **Spectral Variance**. The mean variance across all spectral bands of the hyperspectral image, measuring spectral diversity.

- $H_\lambda$: **Spectral Entropy**. A measure of the information content and complexity across the hyperspectral bands.

- $H_P$: **Panchromatic Entropy**. A measure of the information content and texture in the high-resolution panchromatic (grayscale) image.

- $\sigma_P$: **Panchromatic Standard Deviation**. The standard deviation of the panchromatic image, indicating spatial variability and detail.

- $\max(X)$: The maximum value of a given metric $X$ found across the entire set of scenes being processed. Dividing by this value normalizes each metric to a scale of 0 to 1.

- $S_{R,\text{urban}}$: The adjusted Richness Score for a scene classified as urban.

- $B_U$: The **Urban Boost** factor (e.g., 1.5), a multiplier that increases the score for urban scenes to prioritize their selection.

---

[2]https://prisma.asi.it/js-cat-client-prisma-src/

A.3  NEW VISUALIZATION METHODS FOR DEEPER INSIGHT

To supplement the numerical scores provided by our benchmark, we introduce and standardize a suite of advanced visualization tools. These tools are designed to move beyond simple numerical scores and offer deeper insights into model behavior, particularly regarding performance trade-offs and failure modes. Table A.7 provides a comprehensive summary of these methods. Below, we provide illustrative examples and concrete interpretations based on our experimental findings.

Table A.7: **A Toolkit of New Visualization Methods for Deeper Insight.** Our benchmark moves beyond numerical scores by introducing and standardizing these visualization tools. Each is designed to expose specific performance characteristics and trade-offs that are concealed by traditional, single-score leaderboards.

| Visualization Method | Mechanism | Key Insight Revealed | Contribution to the Benchmark |
|---|---|---|---|
| **Mean Difference Heatmap** | Visualizes the **per-pixel radiometric change** between the input HSI and the final pansharpened output. | Provides an immediate "fingerprint" of a model's fusion strategy, distinguishing between **conservative** (spectrally faithful) and **aggressive** (spatially detailed, but risky) approaches. | Quantifies the **invasiveness** of a fusion strategy and its impact on the original radiometry, moving beyond a single aggregate error number. |
| **VIS vs. Invisible Band Analysis** | Compares model performance on **PAN-overlapping (VIS)** vs. **non-overlapping (SWIR)** bands, typically via a difference map. | Exposes a model's ability to handle the "out-of-band leakage" problem. Reveals if a model is merely copying PAN details or genuinely reconstructing information where no spatial guidance exists. | Provides a crucial diagnostic tool for assessing a model's **scientific reliability**, especially for applications that depend on high-fidelity SWIR data. |
| **False-Color RGB Mapping** | Maps selected HSI channels to a standardized **RGB composite** for intuitive visual inspection. | Compresses high-dimensional data into a human-perceptible format, enabling a quick assessment of spatial structure and the detection of major **color distortions** or artifacts. | Standardizes a key **qualitative sanity check**, ensuring that numerical scores are always grounded in a baseline of visual plausibility and structural integrity. |
| **Multi-Metric Radar Chart** | Projects a model's performance across **multiple criteria** (e.g., different scenes, different metrics) onto a single polar plot. | Instantly reveals a model's performance profile, highlighting its strengths and weaknesses. Clearly distinguishes between **"specialist"** and **"generalist"** models. | Replaces single-score leaderboards with a **holistic performance profile**, revealing trade-offs and discouraging the practice of "metric-gaming." |

A.4  ILLUSTRATIVE EXAMPLES AND ANALYSIS

**VIS vs. Invisible Band Analysis Diagnoses Spectral Fidelity.**  To diagnose a model's performance in spectral regions with weak PAN guidance, we standardize the use of VIS vs. Invisible (e.g., SWIR) band analysis. As shown in Figure 6, we first analyze the input data itself: the visible bands and SWIR bands show distinct spatial responses, particularly for features like waterways,

which is highlighted in the input difference map. We then apply the same analysis to the model's output. A high-fidelity model should not only reconstruct the individual band groups but also preserve the crucial information in their difference. Our analysis confirms that our top-performing models successfully maintain a strong contrast in the output difference map, providing a crucial diagnostic tool for assessing scientific reliability in bands where the PAN offers no guidance.

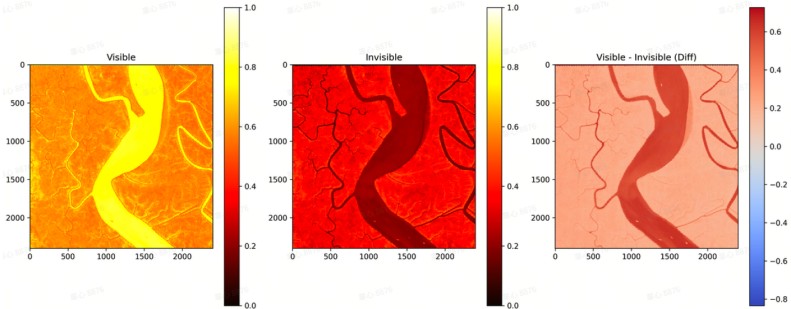

Figure A.7: **False-Color RGB Mapping for Qualitative Assessment.** A false-color composite of the model's output using selected HSI channels. This visualization provides a quick, intuitive check for spatial sharpness, structural integrity, and the absence of major color distortions.

**Standardized False-Color Mapping for Qualitative Sanity Checks.** While simple, a standardized false-color mapping is an essential qualitative check. Our toolbox provides a script to map specific HSI channels (e.g., channels 29, 19, and 0) to an RGB image (Fig. A.7). This compresses the high-dimensional data into a human-perceptible format, allowing for a quick, intuitive assessment of spatial structure and the detection of major color or spectral distortions that might be missed by numerical scores. The varying appearance of different materials in the image reflects the unique spectral sensitivities of the selected channels, providing a useful first-pass analysis.

A.5    USE OF LARGE LANGUAGE MODELS (LLMS)

This document was created with the assistance of a large language model (LLM). The LLM was used to review and refine sentence structure, correct grammatical errors, and improve the clarity of the prose.

