# OpenReview forum: "Re-calibrating Progress: A Physics-Aware Benchmark to Expose the Evaluation Gap in Scientific Machine Learning"
_ICLR.cc/2026/Conference — ICLR 2026 Conference Withdrawn Submission_

### Official Review · Reviewer_ikeP · 2025-10-22

**Soundness:** 2
**Presentation:** 3
**Contribution:** 2
**Rating:** 2
**Confidence:** 4

**Summary:**

The paper motivates an evaluation gap in hyperspectral pansharpening, where models that perform well on standard benchmarks often fail in real-world conditions due to idealized data and fragile metrics. It introduces PRISMABench, a physics-aware benchmarking framework that includes a curated dataset, a new evaluation metric called the PAN-Conditioned Spatial Score, and a visualization toolkit for interpretability. The new metric measures spatial consistency by evaluating only on the spectral range observed by the panchromatic sensor, aligning model evaluation with actual sensor physics.

**Strengths:**

-	Well written introduction and motivation for the evaluation gap in HS-PAN, identifying deficiencies in current benchmarking.
-	The meta-analysis comparing reduced-resolution vs. full-resolution rankings supports the evaluation-gap.
-	Dataset curation is performed via a Richness Score, combining multiple normalized metrics (Eg. Spectral and Panchromatic Entropy), which favors complex, high entropy scenes.
-	The PAN-Conditioned Spatial Score is physically motivated as it explicitly measures spatial fidelity only where the PAN carries information, aligning evaluation with real acquisition and addressing the issue of potentially misleading penalization outside this range.
-	The paper shows that the new metric captures complimentary information to current used metrics by outlining how benchmark methods rank differently.
-	The visualization tools (mean-difference heatmaps, radar charts) provide interpretable insights and complementary perspectives beyond leaderboards.

**Weaknesses:**

-	The PAN-Conditioned Spatial Score only evaluates spatial consistency within the PAN range and does not assess SWIR or other non-overlapping bands, making it insufficient as a standalone HS-PAN quality measure. Beyond this, the authors do not introduce additional quantitative metrics that address the evaluation gap outside the PAN spectrum.
-	The results section and appendix compare the new metric to existing ones (Table 5, Figure 5), show a meta-analysis of results from another paper (Figure 3), and qualitative visual analyses, however the claim that the new PAN-Conditioned Spatial Score helps narrowing the evaluation gap is not sufficiently supported by quantitive results.
-	Inconsistency: Figure 3 reports ρ = 0.49 while the text states ρ = 0.22.
-	The Richness Score is defined but not sufficiently justified by showing e.g. correlation with benchmark difficulty, diversity, or performance.
-	The claim that PRISMABench serves as a blueprint for scientific benchmarking beyond HS-PAN is not supported.
-	The new visualizations and metric address relevant subproblems but lack depth in evaluation, and it is not clear enough outlined how these should be best used for ranking and developing methods in the future.

**Questions:**

-	Please clarify the inconsistency in Figure 3 (ρ = 0.49) vs. text (ρ = 0.22).
-	Based on PRISMABench, how should HS-PAN methods be assessed in the future - using only the new PAN-Conditioned Spatial Score or multiple metrics? How can visual outputs be used for reliable, objective comparison?
-	The new PAN-Conditioned Spatial Score measures spatial consistency within the PAN range but is insufficient for holistic HS-PAN assessment. How do you address spectral and spatial fidelity beyond the PAN range?
-	How do you address data perturbations such as noise, atmospheric effects, or sub-pixel misregistration outlined in Introduction?

---

### Official Review · Reviewer_nVdC · 2025-11-01

**Soundness:** 3
**Presentation:** 3
**Contribution:** 2
**Rating:** 2
**Confidence:** 3

**Summary:**

This paper addresses a critical evaluation gap in scientific machine learning, arguing that models for hyperspectral pansharpening that succeed on idealized, synthetic benchmarks often fail in real-world deployment. To solve this, the authors introduce PRISMABENCH, a new physics-aware evaluation ecosystem. This ecosystem features three main contributions: a large, diverse dataset of 10 real-world PRISMA satellite scenes; a suite of principled metrics, including a novel PAN-Conditioned Spatial Score ($D_{\rho}^{PAN}$) for more robust, no-reference assessment; and insightful visualization tools, like multi-metric radar charts, to expose performance trade-offs that single-score leaderboards hide. Using this new framework, the paper demonstrates a significant disconnect between model rankings on traditional synthetic benchmarks and their actual performance on real-world data, providing a new blueprint to guide the field toward developing more robust and physically plausible models.

**Strengths:**

1. The paper clearly diagnoses the problems of spectral mismatch, synthetic to real gap, and fragile no‑reference metrics, and motivates why HS‑PAN needs physics‑aware evaluation.
2. The authors introduce a PRISMABENCH dataset. This new resource addresses key limitations of prior benchmarks by providing a larger and more diverse collection of 10 globally distributed scenes.
3. The proposal of the PAN-Conditioned Spatial Score ($D_{\rho}^{PAN}$). Instead of naively comparing all bands, this metric is physics-aware in a practical way, focusing the spatial quality assessment only on the spectral bands where the high-resolution PAN sensor actually provides reliable ground-truth spatial information.

**Weaknesses:**

1. The test set expands prior PRISMA FR corpora from 2–4 to 10 scenes, and tiles are larger (2400×2400). That is welcome but still small for a benchmark intended to “re‑calibrate progress.” The benchmark is single satellite (PRISMA only), so conclusions about robustness and real‑world utility may not transfer to other HSI platforms (e.g., differing PAN SRFs/PSFs, swath, radiometry).
2. Because $D_{\rho}^{PAN}$ focuses on PAN-overlapping bands, a method could over-inject PAN structure to improve $D_{\rho}^{PAN}$ while degrading SWIR (non-overlapping) fidelity. The paper mitigates this with VIS-vs-SWIR visual analysis but lacks a quantitative SWIR-specific metric to complement $D_{\rho}^{PAN}$.
3. The paper correctly identifies complex physical challenges like sensor noise, PSF/SRF mismatch, and misregistration as key components of the synthetic-to-real gap. However, the proposed physics-aware solutions do not address these issues. The $D_{\rho}^{PAN}$ metric is only "physics-aware" in the sense that it uses the known spectral range of the PAN sensor. This is a very limited application of physics that ignores the more challenging sensor properties the paper itself brought up.

**Questions:**

1. You correctly identify the out-of-band problem as a fundamental challenge. Why then does your primary metric contribution, $D_{\rho}^{PAN}$, explicitly avoid evaluating this? Why dismiss the SWIR analysis to a qualitative visualization instead of proposing a quantitative metric for spatial fidelity in these, non-overlapping bands?
2. In your related work, you argue that generative models (DDPMs) and self-supervised learning (SSL) are critical and promising paradigms for this task. Why are none of these modern SOTA approaches included in your experimental baselines?

---

### Official Review · Reviewer_dhRs · 2025-11-01

**Soundness:** 2
**Presentation:** 2
**Contribution:** 1
**Rating:** 2
**Confidence:** 4

**Summary:**

This paper introduces PRISMABENCH, a physics-aware benchmark designed to expose the “evaluation gap” in scientific machine learning. The authors focus on the inverse problem of hyperspectral pansharpening and propose three components: (1) a physics-enriched dataset with real PRISMA hyperspectral and panchromatic pairs, (2) extended PAN-centric evaluation metrics including a new physics-consistency score, and (3) visualization tools such as multi-metric radar charts to analyze performance trade-offs. The work aims to encourage benchmarks that better reflect real-world robustness and physical plausibility.

**Strengths:**

- The motivation to address the _evaluation gap_ in scientific machine learning is relevant and timely.

- The inclusion of real sensor data and multiple evaluation metrics shows genuine effort toward practical benchmarking.

**Weaknesses:**

- The **motivation** is not clearly developed — the connection between the “evaluation gap” and the proposed benchmark is abrupt and insufficiently justified.

- The paper reads more like a conceptual or exploratory study than a rigorous machine learning contribution expected at ICLR.

- In Figure 1, visualizations such as radar charts are qualitative and do not necessarily improve evaluation objectivity.

- Terms like PSF and SRF (L80) are unexplained.

- Table 2 lacks background on the selected baseline methods.

- Table 4 and the overall framing make the paper appear outside ICLR’s typical scope, as the audience may not find the problem compelling enough.

- Dataset construction details are missing — it is unclear how the real full-resolution data in Section 5.1 were collected or processed.

- In Figure 5, conclusions about “specialists” and “generalists” seem premature given only 10 scenes.

**Questions:**

Please refer to Weaknesses.

The idea of promoting physics-aware benchmarking is interesting, but the paper lacks rigor, methodological depth, and clarity in motivation and experimental validation. It may be better suited for a remote sensing or applied ML venue rather than ICLR.

---

### Official Review · Reviewer_YxcM · 2025-11-05

**Soundness:** 1
**Presentation:** 1
**Contribution:** 1
**Rating:** 0
**Confidence:** 5

**Summary:**

The paper promises a benchmark framework, termed _PRISMABench_, that evaluates the accuracy of hyperspectral pansharpening based on 10 PRISMA satellite scenes globally distributed introducing a modified metric that restricts statistical evaluation of standard measures for pansharpening to the bandwidth of the panchromatic image. The manuscript introduces a collection of visualizations that support the human evaluation of hyperspectral pansharpening algorithms.

**Strengths:**

I am unfortunately unable to identify any strength from the contents of this submission. I do not want to rule out the option this work was generated by an LLM.

**Weaknesses:**

While reviewing, a major observation with this paper is the gap between claims (in the abstract) versus scientific contents delivered:
* title *A PHYSICS-AWARE BENCHMARK TO EXPOSE THE EVALUATION GAP IN SCIENTIFIC MACHINE LEARNING* is extremly broad for the narrow, domain-specific contribution of the single downstream task of hyperspectral pansharpening of satellite imagery, where the *PHYSICS-AWARE* part reduces to focus on hyperspectral bands that overlap with the spectral range of the panchromatic sensor
* l25-27: code - Where will the reference implementation be made publicly available, and under which license?
  > By open-sourcing our ecosystem, we provide a blueprint for creating benchmarks that challenge the community
* l91: dataset - The total dataset size in bytes / pixels is approx. (scenes x bands x length x width): 10 x 240 x 1000 x 1000 + 10 x 1 x 6000 x 6000 =  2.8G which is not large-scale by today's standards for hyperspectral datasets, e.g. SpectralEarth provides multi-terabyte dataset (for self-supervised pre-training of foundation models)?
  > A Large, Diverse, and Challenging Dataset
* l198: tooling software - According to Appendix A.3, limited novelty beyond standard `matplotlib` visualization that a trained data scientist would intuitively utilize.
  > set of insightful visualization tools designed to move beyond single-score leaderboards and reveal deeper performance trade-offs

Moreover:
- limited readability of main text for general ICLR audience due to the introduction of many hyperspectral remote sensing terms assuming domain expertise, for example:
    * Fig. 2 is hard to grasp without an expert in hyperspectral pansharpening. And even then, what does the row _Ideal_ indicate, pressumably best score? The main text does not provide details either.
    * What about **TV** in l322?
- minor: multiple definitions of abbreviations: *SOTA* (e.g., l212, l302) vs. *SoTA* (e.g., l280-280)
- Fig 1: I understand the cartoon nature, but arrow to right confusing, tiny texts and visualizations limit value, caption: _Left_ / _Right_ vs. 4 quadrants. For a fact, the left bottom table seems a copy-paste of Table VIII in https://doi.org/10.1109/TGRS.2025.3583877 without proper citation.
- Fig. 2: superficial contents
- line 183: cited SatMAE is not even meant as hyperspectral foundation model, cf. HyperSIGMA, SpectralEarth, etc.
- The manuscript does not provide any novel model for pansharpening, it rather benchmarks on existing models, cf. Sect. 5.1.
- Discussion and Conclusions (Sects. 6 & 7) are a stub of generic statements with limited novel insights beyond common sense.

**Questions:**

Tab. 5 suggests that all "traditional" methods favor deep learning models, while the newly introduced metric favors a traditional method. That is a finding which needs strong justification, so why?

---

### Note · Authors · 2025-11-24

I have read and agree with the venue's withdrawal policy on behalf of myself and my co-authors.